# Transcriptomic Studies Suggest a Coincident Role for Apoptosis and Pyroptosis but Not for Autophagic Neuronal Death in TBEV-Infected Human Neuronal/Glial Cells

**DOI:** 10.3390/v13112255

**Published:** 2021-11-10

**Authors:** Mazigh Fares, Kamila Gorna, Noémie Berry, Marielle Cochet-Bernoin, François Piumi, Odile Blanchet, Nadia Haddad, Jennifer Richardson, Muriel Coulpier

**Affiliations:** 1UMR1161 Virologie, Anses, INRAE, Ecole Nationale Vétérinaire d’Alfort, Université Paris-Est, F-94700 Maisons-Alfort, France; Mazigh.Fares@glasgow.ac.uk (M.F.); kamila.gorna@anses.fr (K.G.); noemie.berry@vet-alfort.fr (N.B.); marielle.cochet@vet-alfort.fr (M.C.-B.); francois.piumi@vet-alfort.fr (F.P.); jennifer.richardson@vet-alfort.fr (J.R.); 2MRC-University of Glasgow Centre for Virus Research, Glasgow G61 1QH, UK; 3CHU Angers, Centre de Ressources Biologiques, BB-0033-00038 Angers, France; odblanchet@chu-angers.fr; 4UMR BIPAR 956, Anses, INRAE, Ecole Nationale Vétérinaire d’Alfort, Université Paris-Est, F-94700 Maisons-Alfort, France; nadia.haddad@vet-alfort.fr

**Keywords:** central nervous system, flavivirus, neuronal death, pathological modeling, tick-borne encephalitis virus, TNF family members, regulated cell death

## Abstract

Tick-borne encephalitis virus (TBEV), a member of the *Flaviviridae* family, *Flavivirus* genus, is responsible for neurological symptoms that may cause permanent disability or death. With an incidence on the rise, it is the major arbovirus affecting humans in Central/Northern Europe and North-Eastern Asia. Neuronal death is a critical feature of TBEV infection, yet little is known about the type of death and the molecular mechanisms involved. In this study, we used a recently established pathological model of TBEV infection based on human neuronal/glial cells differentiated from fetal neural progenitors and transcriptomic approaches to tackle this question. We confirmed the occurrence of apoptotic death in these cultures and further showed that genes involved in pyroptotic death were up-regulated, suggesting that this type of death also occurs in TBEV-infected human brain cells. On the contrary, no up-regulation of major autophagic genes was found. Furthermore, we demonstrated an up-regulation of a cluster of genes belonging to the extrinsic apoptotic pathway and revealed the cellular types expressing them. Our results suggest that neuronal death occurs by multiple mechanisms in TBEV-infected human neuronal/glial cells, thus providing a first insight into the molecular pathways that may be involved in neuronal death when the human brain is infected by TBEV.

## 1. Introduction

Tick-borne encephalitis virus (TBEV) belongs to the genus *Flavivirus* (family *Flaviviridae*), whose members include several important human pathogens transmitted by arthropods, such as Japanese encephalitis virus (JEV), West Nile virus (WNV), Zika virus (ZIKV) and Powassan virus (POWV). Most TBEV infections are asymptomatic or cause mild flu-like symptoms, but more severe symptomatic cases also occur with neurological manifestations such as encephalitis, meningo-encephalitis and meningo-encephalomyelitis. These severe conditions, referred to as tick-borne encephalitis (TBE), may lead to death or result in long-term neurological sequelae. Medically speaking, TBEV is the most important arbovirus in Europe and North-Eastern Asia, with 2000 to 4000 human cases per year in the European Union [1] and 8000 to 13,000 cases per year reported worldwide [2]. The incidence of the disease has increased in recent decades, in probable relation to climate change leading to expansion of tick habitats and increase in their abundance [3] and, despite the existence of efficient vaccines, to insufficient coverage in groups at risk. There is currently no therapy available for TBE [4].

TBEV is usually transmitted to humans by infected ticks, mainly of the *Ixodes* family, but may also occasionally be acquired by consumption of unpasteurized dairy products from infected livestock [5,6,7]. In both cases and in certain circumstances that are not as yet well defined, the virus enters the brain in which it causes multiple lesions, including inflammatory changes, neuronal damage and glial reactivity [8,9]. Neuronal damage may be mediated either directly by viral infection, as neurons are the primary target of infection [10], or indirectly by infiltrating immunocompetent cells, inflammatory cytokines and activated resident glial cells such as astrocytes and microglial cells [11,12].

The molecular mechanisms by which neurons die upon TBEV infection are still largely unknown. Several processes may be involved as multiple types of neuronal death have been uncovered in the last decades. These include both non-regulated cell death (necrosis) and regulated cell death such as pyroptosis, apoptosis or autophagic death [13]. Apoptosis, the best known of the regulated cell death types, has been described in several neurological diseases [14,15,16,17,18], including in brain infection [19,20]. TBEV-induced apoptosis has been reported in neurons in the brains of mice and monkeys [21,22] as well as in human neuroblastoma cell line [23] and more recently, in human neurons differentiated from cultured fetal neural progenitor cells [24], suggesting it may play a role in TBE. It has; however, not as yet been observed in human brains of infected patients [9]. Pyroptosis is a more recently discovered regulated form of necrosis [13]. It leads to the formation of pores in the plasma membrane that disrupt membrane integrity and allow the release of pro-inflammatory factors [25]. It is thus an inflammatory process, as opposed to apoptosis. Pyroptosis has also been implicated in several neurological diseases [26,27] and certain neurotropic viral infections [28,29,30], although evidence for its involvement in the central nervous system is scarce for viral infection in general and inexistent for TBEV infection. Autophagy, although mainly known for providing the nutrients necessary for cell survival, is also thought to lead to cell death when excessively activated [31]. Its activation upon infection by several flaviviruses, including DENV [32,33], WNV [34], ZIKV [35] and TBEV [36], suggested that it may play a role in flaviviral-induced neuronal death.

Recently, we set up and characterized a new in vitro model of TBEV infection of the central nervous system (CNS) using complex cultures of human neuronal/glial cells derived from fetal neural progenitors. In these cultures, we showed that TBEV infection mimics several hallmarks of in vivo infection, including marked neuronal tropism and neuronal death [24], thus providing a unique model for investigating the molecular mechanisms by which TBEV infection leads to neuronal death in the human brain. Here, using this model, we sought for evidence of different types of cell death. Our results confirmed apoptotic death and revealed an up-regulation of genes that suggest extrinsic and intrinsic apoptosis as well as pyroptosis but not autophagic death. These results suggest that neuronal death occurs by multiple mechanisms in TBEV-infected human neuronal/glial cells and provide leads for identification of important molecular pathways in TBE.

## 2. Materials and Methods

### 2.1. Ethics Statement

Human fetuses were obtained after legal abortion with written informed consent from the patient. The procedure for the procurement and use of CNS tissue from the human fetus was approved and monitored by the “Comité Consultatif de Protection des Personnes dans la Recherche Biomédicale” of Henri Mondor Hospital, France. The cells are declared at the Research Ministry of France. Reference numbers are AC-2017-2993 (“Centre de Ressources Biologiques”, University Hospital in Angers BB-0033-00038) and DC-2019-3771 (UMR Virology).

### 2.2. Culture of Human Neural Progenitor Cells

Human neural progenitor cells (hNPCs) were prepared and cultured as previously described in [37,38].

### 2.3. Neuronal and Glial Differentiation

Mixed cultures of neuronal/glial cells were obtained by differentiation of hNPCs as previously described in [24]. Ninety-six-well plates (Greiner, 655090, Dutscher, Bernolsheim, France) were used for fluorescent immunostaining.

### 2.4. Magnetic-Activated Cell Sorting

Cultures enriched for neurons (En-Ne) or astrocytes (En-As) were obtained as previously described in [24]. Cells that had been trypsinized but not sorted for enrichment were named unsorted cells (Uns-C).

### 2.5. Virus and Infection

TBEV Hypr strain was a kind gift from S. Moutailler (Maisons-Alfort, France). The strain was isolated in 1953 from the blood of a 10-year-old child in the Czech Republic and the complete sequence was published in [39]. Virus amplification, titration and infection (MOI 10^−2^) were as described previously in [24]. All procedures with infectious materials were performed under bio-safety level-3 (BSL-3) conditions.

### 2.6. RNA Isolation and qPCR

RNA was isolated from infected and non-infected neuronal/glial cells and in cultures enriched in neurons (En-Ne) or astrocytes (En_As). Cells were lysed using the NucleoMag^®^ 96 RNA kit (Macherey Nagel, Hoerdt, France) and RNA was extracted with a King Fisher Duo automat (Fisher Scientific, Illkirch, France) following the manufacturer’s instructions. Two hundred and fifty ng (cultures without enrichment) or 150 ng (cultures with enrichment) of RNA were used to synthesize cDNA with the SuperScript™ II Reverse Transcriptase kit (Thermo Fisher Scientific, Illkirch, France). Real-time PCR was performed using 2 µL of cDNA and QuantiTect SYBR green PCR master (Qiagen, Courtaboeuf, France) with a LightCycler 96 instrument (Roche Applied Science, Meylan, France), for a total volume of 20 µL of reaction mixture. For relative quantification, the −2^ΔΔCt^ method was used [40]. The reference gene was *HPRT1*. Primers pairs are listed in Appendix A. * indicates that primers were designed using “primer designer” from the website: https://www.bioinformatics.nl/cgi-bin/primer3plus/primer3plus.cgi, accessed on 16 August 2021. PCR efficiency determined for each pair of primers was greater than 97% with the exception of TNFSF10 (=91%).

### 2.7. RT^2^ Profiler PCR Array

Equal volumes of RNA from biological triplicates were pooled for each condition. Five hundred ng of RNA were transcribed with the RT^2^ First Strand Kit (SA Biosciences, Qiagen, Courtaboeuf, France). Synthetized cDNA was subjected to human PCR arrays related to apoptosis and autophagy (RT^2^ Profiler PCR array—PAHS-012Z and PAHS-084Z SA Biosciences, Qiagen, Courtaboeuf, France), according to the manufacturer’s instructions. Data were normalized using the *HPRT1* house-keeping gene and analyzed with the −2^ΔΔCt^ method for relative quantification. According to the manufacturer’s instructions, an arbitrary cut-off of 3 was applied to determine significant differences. The analysis was performed using the Qiagen Data analysis center (http://www.qiagen.com/fr/shop/genes-and-pathways/data-analysis-center-overview-page/, accessed on 15 May 2017).

### 2.8. Immunofluorescence Assays and Cell Enumeration

Neuronal/glial cells were fixed for 30 min in 4% paraformaldehyde in PBS (Electron Microscopy Sciences, Euromedex, Souffelweyersheim, France) and standard immunofluorescence was performed using antibodies for cleaved-caspase 3 (Cell Signaling Technology, 9661, Ozyme, St. Quentin, France) and double-stranded RNA (Scicons, K1, Szirak, Hungary). Cells were blocked for 2 h in 3% BSA (Sigma, St. Quentin Fallavier, France), 0.3% Triton-X-100 (VWR, Rosny-sous-Bois, France) in PBS and primary antibody was incubated in 1% BSA, 0.1% Triton-X-100 in PBS overnight at +4 °C. Secondary antibody was Alexa Fluor-488-conjugated anti-rabbit IgG (Molecular Probes, Invitrogen, Thermo Fisher Scientific, Illkirch, France). Nuclei were stained with 4′,6-diamidino-2-phenylindole (DAPI) (Life Technologies, Thermo Fisher Scientific, Illkirch, France) at 0.1 ng/mL. Immuno-stained cells were enumerated manually. Images were acquired with an AxioObserver Z1 (Zeiss) inverted microscope using ZEN software (Zeiss) and analyzed using ImageJ 1.49 m software. An average of 1200 cells per well were enumerated per replicate. The digitized images shown were adjusted for brightness and contrast using ZEN.

### 2.9. Statistical Analyses

Data are represented as mean ± standard deviation (SD). Statistical analyses were performed with GraphPad Prism V4.03 or V6.0.1 using an unpaired Student’s *t*-test or a one-way ANOVA analysis (Bonferroni’s Multiple Comparison Test), * = (*p* < 0.05), ** = (*p* < 0.01), *** = (*p* < 0.001), non-significant (ns) = (*p* > 0.05).

## 3. Results

### 3.1. Evidence of Apoptotic Death in Human Neuronal/Glial Cells Infected with TBEV

We previously demonstrated that neuronal death occurs in TBEV-infected human neuronal/glial cells [24]. In a time-course study, and under our experimental conditions, death occurred from 72 hpi on (25% neuronal loss) and increased throughout the entire course of infection, reaching 60% of neuronal loss at 7 dpi. We showed that it was accompanied by an increase in TUNEL staining, suggesting that neuronal death occurred through an apoptotic pathway. However, positive TUNEL staining, which reveals fragmented DNA, may also indicate other types of cell death [41,42]. To verify the occurrence of apoptosis, we set up the same experimental design as in [24] (Figure 1A) and immunostained cells with an antibody specific for cleaved-caspase 3 (C3A), a caspase that is central to apoptotic death. At 7 dpi, we observed an increase in C3A staining in TBEV-infected cells compared with their matched non-infected controls (Figure 1B). The staining co-localized with nuclei presenting a damaged morphology (small rounded nuclei). This was confirmed by enumeration of C3A-positive cells at both 72 hpi and 7 dpi (Figure 1C). This increase paralleled the increase in neuronal death previously observed in [24]. On the contrary, no difference was observed between infected and non-infected cells at an earlier time point, 14 hpi (Figure 1C), when neuronal death was not yet observed. Thus, these results confirmed that apoptotic events occur in TBEV-infected human brain cells in culture. To determine whether apoptotic death was due to direct viral infection, we then co-immunostained cells with the C3A antibody and an antibody specific to double stranded viral RNA (dsRNA) that is formed during viral replication. DsRNA staining was observed in all C3A-positive cells at 7 days post-infection (Figure 1D), suggesting that viral infection directly induces apoptosis. Of note, other cells with a damaged nucleus were not stained with C3A antibody and were either stained or not with dsRNA antibody, suggesting the existence of a C3-independent death that can be due to both direct and indirect infection.

### 3.2. Transcriptomic Analyses Reveal Up-Regulation of Genes Involved in Apoptosis and Pyroptosis in TBEV-Infected Human Neuronal/Glial Cells

To gain further insight into the molecular pathways involved in TBEV-induced neuronal death, we used a PCR array approach to analyze the differential expression of 84 human genes known to be involved in apoptotic pathways. As neuronal death was first observed at 72 hpi in our culture conditions [24], transcripts from cells infected with TBEV for 72 h were pooled from biological triplicates and compared with transcripts from their matched non-infected controls. The studied genes are shown in Figure 2A and Appendix A. After applying an arbitrary cut-off of threefold, 16 genes were shown to be significantly modulated in TBEV-infected cells, among which 15 genes were up-regulated and 1 was down-regulated (Figure 2A and Appendix A). Among these were six members of the tumor necrosis factor alpha (TNFα) family and their receptors, TNFSF10, TNFRSF9, TNFα, CD40, TNFRSF1B (TNFR2) and CD70, six members of the caspase family, CASP1, CASP4, CASP8, CASP5, CASP14 and CASP7 and four members of the Bcl2 family, BCL2A1, BIRC3, HRK and BCL2L10. PCR array data were confirmed for six genes, TNFSF10 (Figure 2B), TNFRSF10A (Figure 2C), TNFRSF10B (Figure 2D), TNFα (Figure 2E), TNFRSF1A (Figure 2F) and CASP1 (Figure 2G), using RT-qPCR, attesting to the reliability of the PCR array data. The most up-regulated gene was TNFSF10. A kinetic analysis further revealed that its expression increased as early as 14 hpi and that it remained high up to 7 dpi (fold change of 564 and 554, respectively). TNFSF10 has two death receptors, TNFRSF10A (TRAILR1) and TNFRSF10B (TRAILR2) that are known to be involved in the induction of apoptosis. The PCR array data indicated that they were slightly up-regulated at 72 hpi, although with a fold change lower than 3 (Figure 2A and Appendix A). To gain further insight into their regulation during the course of infection, we also performed RT-qPCRs from 14 hpi to 7 dpi. This showed that the TNFRSF10A gene was slightly up-regulated at 72 hpi and had a more pronounced up-regulation at 7 dpi (fold change of 2.8 and 8.9, respectively). On the contrary, no modulation was observed at the earlier time point of 14 hpi and 24 hpi (Figure 2C). Similarly, up-regulation was also shown for TNFRSF10B from 72 hpi on (fold change of 3.1 and 2.6 at 72 hpi and 7 dpi, respectively) (Figure 2D). Like TNFSF10, TNFα is known as an inducer of apoptosis. The two ligands are involved in the intrinsic pathway of apoptosis. They both activate caspase-8 (CASP8), a caspase whose expression is also up-regulated in TBEV-infected human neuronal/glial cells (Figure 2A and Appendix A, fold change of 5.86 at 72 hpi). We performed RT-qPCR to analyze the kinetics of TNFα expression as well as the kinetics of expression of its death receptor TNFRSF1A (TNFR1). Similar to TNFSF10, TNFα was up-regulated as early as 14 hpi (fold change of 15) and was strongly increased at 7 dpi (fold change of 194) (Figure 2E). According to the PCR array data, TNFRSF1A was expressed by the neuronal/glial cells but it was not modulated by TBEV infection (Figure 2A and Appendix A). RT-qPCR data showed a very slight modulation, with up-regulation lower than 2 at every time point studied (fold change of 1.72 at 7 dpi) (Figure 2F). Thus, we showed that, in TBEV-infected human neuronal/glial cells, two TNF family ligands known to be involved in the extrinsic pathway of apoptotic cell death were strongly up-regulated and their death receptors were either expressed (TNFRSF1) or up-regulated (TNFRSF10A/B) from 72 hpi on. Our results also showed that three pro-apoptotic members of the BCl2 family known to be involved in the intrinsic pathway of apoptosis, BCl2A1, BCl2L10 and HRK, were up-regulated (Figure 2A and Appendix A). This led us to hypothesize that neuronal death in TBEV-infected human neuronal/glial cells may be mediated by both the extrinsic and intrinsic pathways of apoptosis. The second most up-regulated gene as shown by the PCR array data was caspase-1 (CASP1) (Figure 2A and Appendix A). The kinetics analyses by RT-qPCR further revealed that up-regulation occurred from 14 hpi to 7 dpi (fold change of 26 and 70, respectively) (Figure 2G). Of note, not only CASP1 but also caspase-4 (CASP4) and caspase-5 (CASP5), all of which involved in pyroptotic cell death [43], were up-regulated (Figure 2A and Appendix A), suggesting a potential implication of this mode of cell death. To address this possibility, we sought evidence for modulation of gasdermin D (GSDMD) and aim-2 (AIM2), two major pyroptotic genes (44–46). Indeed, kinetic analyses by RT-qPCR showed that GSDMD was up-regulated at every time point studied, with an increase from 14 hpi to 7 dpi (fold change of 9.8 and 33.5, respectively) (Figure 2H), whereas AIM2 was up-regulated from 72 hpi on (fold change of 4.2 and 7.7, at 72 hpi and 7 dpi, respectively) (Figure 2I), further suggesting that pyroptotic pathways occur in TBEV-infected human brain cells.

### 3.3. Transcriptomic Analyses Do Not Show Evidence of Activation of Autophagic Pathway in TBEV-Infected Human Neuronal/Glial Cells

As autophagic pathway is thought to play a role in viral-induced cell death, we also sought evidence for activation of the autophagic pathway in TBEV-infected human neuronal/glial cells. Following the experimental procedure previously described and using the same transcripts (72 hpi), we performed RT-qPCR using a commercial PCR array allowing the analysis of 84 human genes known to be involved in the autophagic pathway. The studied genes are shown in Figure 3A and Appendix A. After applying the arbitrary cut-off of threefold, seven genes were shown to be significantly modulated in TBEV-infected cells, all of them being up-regulated. Again, TNFSF10 was shown to be strongly up-regulated. Of note, six out of the seven up-regulated genes were regulators of both apoptotic and autophagic pathways (TNFSF10, TNFα, IGF1, TGM2, CASP8, and NFkB1), whereas none of them were components restricted to the autophagic machinery, such as implicated in autophagic vacuole formation, vacuole targeting, protein transport, autophagosome-lysosome linkage, ubiquitination and proteases. To confirm the absence of regulation of major genes involved in the autophagic machinery, we verified the expression of two key genes, BECN1 (involved in autophagic vacuole formation) and ATG3 (involved in the ubiquitination process) during the course of infection. The two genes were not modulated at any time point from 14 hpi to 7 dpi (Figure 3B,C), confirming the PCR array data (Figure 3A). This argues that the autophagic pathway does not play a major role in TBEV-induced neuronal death in human neuronal/glial cells.

### 3.4. Human Neurons and Human Astrocytes Differentially Expressed the TNFSF10, TNFRSF10A and TNFRSF10B Genes in Basal and TBEV-Infected Conditions

The up-regulation of TNFSF10 and its two death receptors suggested that they are involved in neuronal death in TBEV-infected human neuronal/glial cells. To increase our understanding of their role, we sought to determine in which particular cell types they are expressed and up-regulated. We used cultures enriched in either neurons (En-Ne) or astrocytes (En-As), which comprise 94.1 +/− 0.4% and 35.7 +/− 2.8% of neurons, respectively and 3.1 +/− 0.4% and 53.5 +/− 2.7% of astrocytes, respectively, as compared with 74.1 +/− 4.1% of neurons and 20.8 +/− 4.9% of astrocytes in mixed human neuronal/glial cultures (previously described in [24]). En-Ne and En-As as well as unsorted neuronal/glial cells (Uns-C) were infected with TBEV for 72 h. This was the latest possible time point, since neurons massively die in En-Ne shortly afterwards [24]. At 72 hpi, the expression of TNFSF10 and its receptors TNFRSF10A/B was evaluated by RT-qPCR in non-infected (basal gene expression) and infected conditions. In basal conditions, TNFSF10 and TNFRSF10B were expressed similarly in En-Ne and En-As cultures (Figure 4A,C) whereas TNFRSF10A was slightly overexpressed in En-As (fold change of 2.5) (Figure 4B). In TBEV-infected cultures, TNFSF10 was up-regulated in both En-Ne and En-As but, whereas up-regulation was moderate in En-Ne (fold change = 7.6), it was massive in En-As (fold change of 690) (Figure 4D), suggesting that astrocytes are the main producers of TNFSF10 in TBEV-infected human neuronal/glial cells. TNFRSF10A displayed a 3-fold up-regulation exclusively in En-As cultures (Figure 4E), whereas TNFRSF10B presented a 3.2-fold up-regulation only in En-Ne cultures (Figure 4F), which suggests a role for TNFSF10 on neurons mediated by TNFRSF10B.

## 4. Discussion

While the mosquito-borne flaviviruses have attracted most of the attention over the last 20 years, tick-borne flaviviruses, such as TBEV, have been far less studied. In both cases, however, very little is known about the molecular mechanisms involved in neuronal death, especially in the human species, although this is a major pathological consequence of these viruses. In this regard, the lack of a physiologically relevant in vitro model has precluded such studies for many years. Here, in a human neuronal/glial cell-based pathological model of TBEV infection, we showed that neuronal death is associated with up-regulation of genes involved in apoptosis and pyroptosis, suggesting that at least two types of death occur coincidently.

It is now considered that neurons, like other cells, may die in multiple manners, including apoptosis, pyroptosis, necroptosis or autophagy, as evidenced in several neurodegenerative diseases [13]. Of note, it was observed that some of the different types of death may be coincident, and that pharmacological inhibition of a single type was insufficient to allow protection of neurons “primed to death”. Apoptosis is the longest known regulated form of cell death and the most studied. It is involved in neuronal death in several pathological conditions [13], including during infection by neurotropic viruses [19,20], among which is TBEV [21,22,44,45]. In this study, and in a previous one [24], we also observed apoptotic events in TBEV-infected neuronal/glial cells, providing further evidence that apoptosis occurs in brain cells upon TBEV infection. During viral infection, neuronal apoptosis may result from activation of intrinsic factors as a direct effect of the virus, or from extrinsic neurotoxic factors, such as members of the TNF family ligands, produced by cells in the immediate environment of the neuron. In an attempt to determine which specific molecular pathways were implicated, we performed transcriptomic studies that revealed up-regulation of genes involved in both intrinsic (BCL2A1, BCL2L10, HRK) and extrinsic (TNFSF10 and its death receptors TNFRSF10A/10B, TNFα and CASP8) pathways, suggesting that both contribute to neuronal death. Involvement of these two apoptotic pathways has already been shown for WNV [44,45]. TNFα and TNFSF10 both play pleiotropic roles in the CNS. TNFα, although best known as inducing apoptosis in a TNFRSF1A-dependent manner, has a particularly complex role in that it possesses not only neurotoxic but also neuroprotective capacities. This has been shown in diverse situations, including upon WNV infection [46,47,48,49,50]. TNFSF10 is a more classic inducer of neuronal apoptosis whose action has been demonstrated in vitro [51,52] and in vivo following auto immune neuro-inflammation [53], ischemia [54] and HIV infection [55]. Whereas it is harmless to the brain in non-pathological conditions, neuronal damage is induced following diverse neurological insults through its up-regulation and up-regulation of its death receptors [56]. The up-regulation of both TNFSF10 in astrocytes and TNFRSF10B in neurons in TBEV-infected human neuronal/glial cells thus suggests that astrocytes may be deleterious for neurons through a TNFSF10/TNFRSF10B-mediated mechanisms. A similar pattern has been shown in other pathological situations such as in the advanced stage of Alzheimer’s disease, where TNFSF10 secreted by astrocytes binds to *TNFRSF10B* on neurons and triggers caspase-8-dependent apoptosis [57]. In our model, however, the increase in TNFSF10/TNFRSF10B at 72 hpi is correlated with only a very small number of cleaved-caspase-3-positive cells, suggesting that other mechanisms may be involved in TBEV-induced death. In addition, a pathological role for astrocyte-derived TNFSF10 is not supported by our previous results, which showed that astrocytes were protective rather than detrimental to neurons in TBEV-infected human neuronal/glial cells [24]. Thus, the exact roles of the members of the TNF family in TBEV-induced neuronal death will have to be fully elucidated in further studies.

Recently, it was suggested that apoptosis may not be the only type of neuronal death occurring in the CNS following *flavivirus* infection. Genes associated with pyroptosis and necroptosis have indeed been shown to be up-regulated in the brain of WNV-infected young mice [28], and necroptosis was shown to occur in JEV-infected murine neurons [58]. However, evidence for these types of cell death is scarce and requires validation by further studies. Of note, apoptosis was not observed in the brain of human patients infected with TBEV [9], indicating that this is probably not the most important type of TBEV-induced neuronal death. In addition, in in vitro [23] and in vivo [12,21,22] models of TBEV infection, necrosis has often been described together with apoptosis, but whether regulated forms of necrosis played a role has so far not been investigated. In favor of the existence of non-apoptotic death in TBEV-infected human neuronal/glial cells is our observation of a strong difference in the percentage of TUNEL staining and cleaved-caspase 3 staining at 72 hpi, the first one being 10 times greater than the second ([24] and results from this study). As TUNEL staining reveals fragmented DNA, a process that occurs not only in apoptosis but also in other types of cell death, including pyroptosis [41,42], it is likely that this difference reflects multiple pathways leading to DNA fragmentation and death. Furthermore, our results showed an up-regulation of two major pyroptotic factors, GSDMD and AIM2, as well as of CASP1 and CASP4/5, involved in the canonical and non-canonical pyroptotic pathways [59], providing evidence for the first time that this regulated form of necrosis may be another mode of neuronal death occurring upon TBEV infection. This finding strengthens recent studies by others suggesting a role for pyroptosis in flavivirus-induced death in the CNS [28,56,60,61]. Necroptosis is another type of regulated necrosis. It may be triggered by up-regulation of TNF family ligands when apoptotic events are prevented, in the case of cells lacking CASP8 [62]. Whether necroptotic pathways are activated in TBEV-infected human neuronal/glial cells remains to be evaluated, but it would be unlikely to be mediated via the TNF family ligands as CASP8 is up-regulated in these cells.

Although the role of autophagy in cell death is still under debate, and there is conflicting literature about whether autophagy is simply associated with other types of cell death or represents a distinct cell death process [31,63], report of its activation upon infection by several viruses including flaviviruses such as DENV [64], WNV [34], ZIKV [35] and TBEV [36] prompted us to evaluate its role in TBEV-infected human neuronal/glial cells. Although we analyzed the expression of 84 human genes involved in autophagy, we did not find strong evidence in favor of its involvement. Indeed, major genes of autophagy, like BECN1 or ATG3, were not modulated and most of the genes we found up-regulated were not specific to autophagy but rather shared by both apoptotic and autophagic pathways. Our result is in apparent contradiction with the study by Bily et al. [36], which showed that autophagy was activated in human neuroblastoma cells and that it enhanced TBEV replication. This divergence may be due to differences in the methodology or the nature of the cells used in the two studies.

Understanding neuronal death following infection with TBEV and other neurotropic flaviviruses is critical as this will allow the rational design of therapeutic molecules with neuroprotective capacities. Indeed, it will probably be necessary to combine drugs with antiviral and neuroprotective properties to not only cure viral infection, but also limit the neurological sequelae that are often observed following viral encephalitis. In this study, using TBEV-infected human neuronal/glial cells, we provide the first evidence suggesting that two types of regulated neuronal death, apoptosis and pyroptosis, concur to damage brain cells. This advances our understanding of TBEV-induced neuronal death and provides a model for future functional studies that will be needed to confirm the role of apoptotic and pyroptotic pathways, determine whether other types of neuronal death occur, define their relative contribution and, finally, decipher the precise molecular pathways involved.

## Figures and Tables

**Figure 1 viruses-13-02255-f001:**
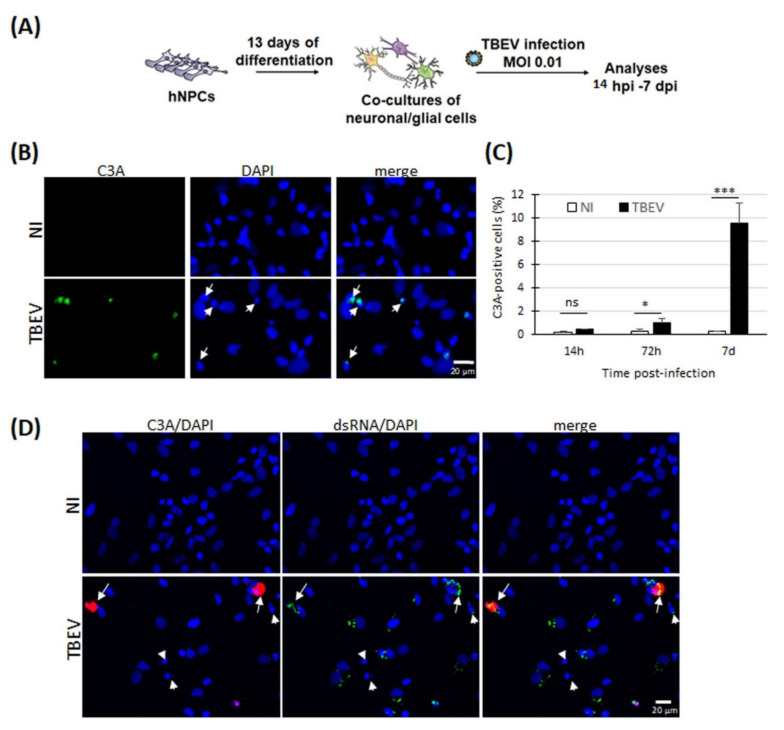
TBEV induced apoptotic death in human neuronal/glial cells. (**A**) Schematic representation of the experimental procedure. (**B**) Immunofluorescence labeling of human neuronal/glial cells 7 days following TBEV infection. An antibody directed against cleaved-caspase-3 (C3A) revealed apoptotic cells. Nuclei were stained with DAPI (blue). Note the co-localization of cleaved-caspase-3 staining and apoptotic nuclei (small, rounded nuclei). (**C**) Enumeration of apoptotic cells based on immunofluorescence labeling of C3A. (**D**) Immunofluorescence labeling of human neuronal/glial cells 7 days following TBEV infection. Cleaved-caspase-3 (C3A) and dsRNA antibodies revealed apoptotic and infected cells, respectively. Nuclei were stained with DAPI (blue). Arrows indicate co-localization of C3A and dsRNA staining, associated with damaged nuclei. Arrowheads indicate damaged nuclei without C3A and with or without dsRNA staining. Results are representative of 3 independent experiments performed in triplicate. Data are expressed as the mean ± SD. Statistical analysis was performed using one-way ANOVA (Bonferroni’s Multiple Comparison Test) with GraphPad Prism V. 6.0.1, ns = non-significant (*p* > 0.05); * = *p* < 0.1, *** = *p* < 0.001. Scale bar = 20 µm.

**Figure 2 viruses-13-02255-f002:**
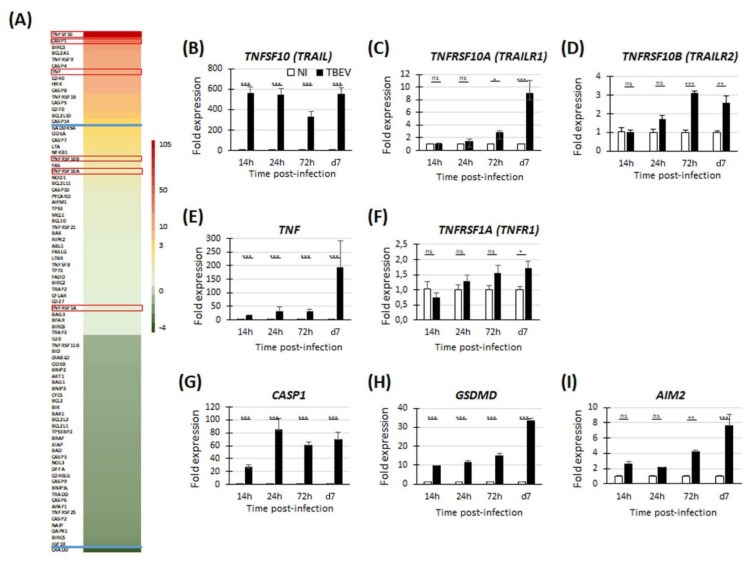
TBEV induced apoptosis and pyroptosis in human neuronal/glial cells. (**A**) TBEV-infected neuronal/glial cells and their matched non-infected controls were analyzed 72 hpi using an RT^2^ Profiler PCR array specific for human apoptotic signaling. The heat map shows the differential expression of 84 analyzed human genes. The most highly up- and down-regulated genes are colored in red and dark green, respectively. The blue lines indicate the arbitrary cut-off of 3. Genes between the two lines are considered non-regulated. (**B**–**I**) RT-qPCR analyses of selected genes (framed in red in (**A**)) from 14 hpi to 7 dpi. Gene expression was normalized to the *HPRT1* gene and the −2^ΔΔCt^ method was used for relative quantification (compared with non-infected cells at the same time point). Data are expressed as the mean ± SD. Results are representative of at least two independent experiments performed in triplicate. Statistical analysis was performed using a two-tailed unpaired *t*-test with GraphPad Prism V6.0.1, ns = non-significant (*p* > 0.05); * = *p* < 0.05; ** = *p* < 0.01; *** = *p* < 0.001.

**Figure 3 viruses-13-02255-f003:**
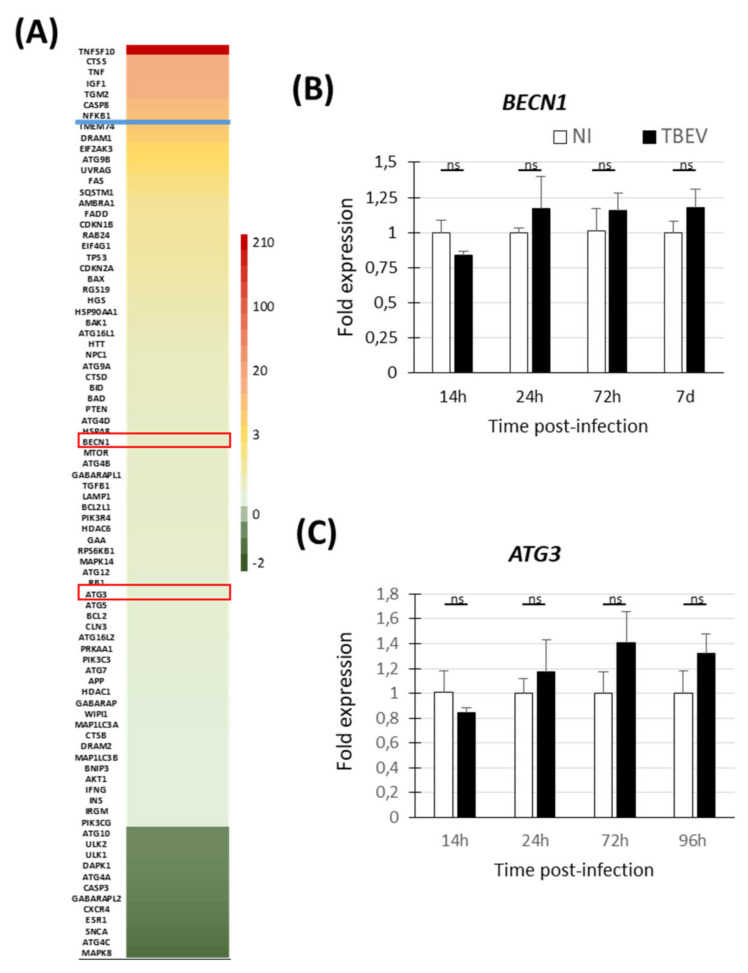
No evidence of autophagy in TBEV-infected human neuronal/glial cells. (**A**) TBEV-infected human neuronal/glial cells and their matched non-infected controls were analyzed 72 hpi using an RT^2^ profiler PCR array specific for the autophagic pathway. Heat map showing the 84 human genes analyzed and their differential expression. Color code and blue line are as in Figure 2. (**B**,**C**) RT-qPCR analyses of selected antiviral response genes (framed in red in Figure 2A). Gene expression was normalized to the HPRT1 gene and the −2^ΔΔCt^ method was used for relative quantification (compared with non-infected cells at the same time point). The results are expressed as the mean ± SD. They are representative of two independent experiments performed in triplicate (**B**,**C**) and one experiment performed with pooled triplicates (**A**). Statistical analyses were performed with GraphPad Prism V6.0.1 using a two-tailed unpaired *t*-test, ns = non-significant (*p* > 0.05).

**Figure 4 viruses-13-02255-f004:**
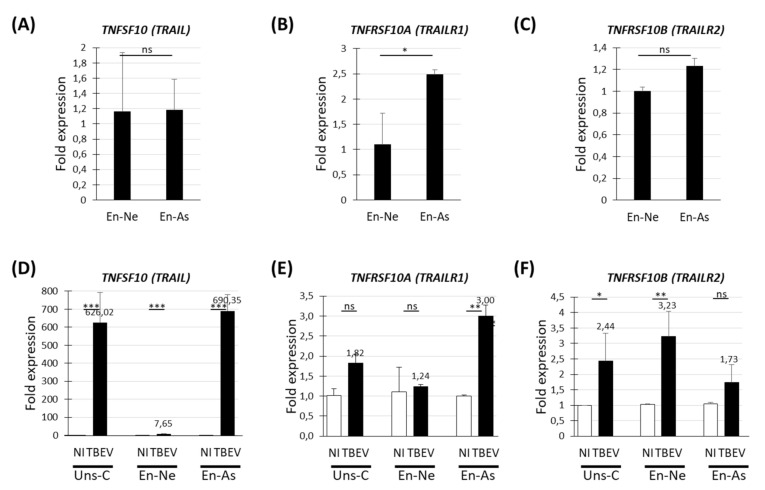
Differential expression of TRAIL and its death receptor genes in cultures enriched in either human neurons or human astrocytes. HNPC-derived neuronal/glial cells differentiated for 13 days were either replated as co-cultures (Uns-C) or sorted using MACS technology and enriched in neurons (En-Ne) or astrocytes (En-As). (**A**–**F**) RT-qPCR analyses of *TRAIL* and its two death receptors genes. (**A**–**C**) RT-qPCR analyses. Basal gene expression in En-As vs. En-Ne cultures. (**D**–**F**) RT-qPCR analyses. TBEV-induced gene expression in the 3 cultures. The results are expressed as the mean ± SD. Data are representative of two independent experiments performed in triplicate. Statistical analyses were performed with GraphPad Prism V6.0.1 using a two-tailed unpaired *t*-test, ns = non-significant (*p* > 0.05); * = *p* < 0.05; ** = *p* < 0.01; *** = *p* < 0.001.

## Data Availability

Data supporting the conclusions of this work have been presented in the manuscript.

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
