# Peer review of "Transcriptomic Studies Suggest a Coincident Role for Apoptosis and Pyroptosis but Not for Autophagic Neuronal Death in TBEV-Infected Human Neuronal/Glial Cells"

_viruses, 2021, doi:10.3390/v13112255_

Round 1
Reviewer 1 Report
Dear Authors,
The title describes the goal of the study quite well. It is whether transcriptome analysis supports one of 3 pathways of regulated cell-death in neuron/glia (mixed) cell culture model. I am not sure if it is a representative model, and if it correlates with immunohistochemistry/ in situ hybridization findings on human brain tissue. I am concerned that only one MOI (10ee-2)was chosen, and all that was observed here is an artevact? I am not sure how established this model is; however, I know that viral genome is typically detected in lowest abundance. It might have been beneficial if variable MOIs were chosen in case this model is not well established.
Author Response
Pleease, see the attachment

Reviewer 2 Report
The authors examined the mechanism of TBEV-induced cell death such as apoptosis, pyroptosis, or autophagy-related cell death by gene expression analysis in this study. The study is interesting, but I would like the authors to consider the following unclear points.
Major comments
1. In figure 1 showed the rate of cleaved-caspase 3 positive apoptotic cells. The authors also showed the possibility of pyroptosis in figure 2. These results indicated that TBEV infection in hNPCs-derived neuronal cells induce many types of cell death. Thus, the authors should show the total number of death cells induced by viral infection, using such as PI staining, and should show the rate of the number of apoptotic cells in whole dead cells.
2. It should be shown that TBEV infection induces apoptosis related to caspase 3, such as double staining for cleaved-caspase 3 and viral antigen.
3. Since figure 2 showed the possibility of pyroptosis, the occurrence of pyroptosis should be shown by detecting the cleavage of GSDMD by such as immunoblotting.
4. The reason why the percentage of caspase 3-positive cells was not so high (about 1 %) after 72 hpi in figure 1, even though the expression of apoptosis-inducing factors was significantly increased in figure 4, should be discussed. These results suggested that other types of cell death (such as pyroptosis) may be more important than apoptosis in these cells.
Minor comments
1. The authors should correct “J7” to “7d” in figures 2B to 2I.
2. In line 102, the authors should correct “3.4.” to “2.4.”.
3. There should be a space before a unit such as ul, h, or hpi.
Author Response
Please, see the attachment .
Best regards,
MC

Round 2
Reviewer 1 Report
Thank you for your explanations.
Reviewer 2 Report
Undetectable for the cleaved GSDMD was disappointing, but all comments were adequately answered.